# Beyond Structured Attributes: Image-Based Predictive Trends for Chest X-Ray Classification

**Katharina Hoebel**[*1]             KHOEBEL@DS.DFCI.HARVARD.EDU
[1] *Dana-Farber Cancer Institute*
**Jesseba Fernando**[*2]            FERNANDO.JE@NORTHEASTERN.EDU
[2] *Network Science Institute, Northeastern University*
**William Lotter**[1]              LOTTERB@DS.DFCI.HARVARD.EDU

**Editors:** Accepted for publication at MIDL 2024

## Abstract

A commonly emphasized challenge in medical AI is the drop in performance when testing on data from institutions other than those used for training. However, even if models trained on distinct datasets perform similarly well overall, they may still exhibit other systematic differences. Here, we study these potential dataset-centric prediction variations using two popular chest x-ray datasets, CheXpert (CXP) and MIMIC-CXR (MMC). While CXP-trained models generally perform better on CXP than MMC test data and vice versa, this performance decrease is not uniform across individual images. We find that image-level variations in predictions are not random but can be inferred well above chance, even for pathologies where the overall performance gap is small, suggesting that there are systematic tendencies of models trained on different datasets. Furthermore, these "predictive tendencies" are not solely explained by image statistics or attributes like radiographic position or patient sex, but rather are pathology-specific and related to higher-order image characteristics. Our findings stress the complexity of AI robustness and generalization, highlighting the need for a nuanced approach that especially considers the diversity of pathology presentation.

**Keywords:** Chest x-ray classifier, domain shift, generalization, image statistics

## 1. Introduction

Deep learning (DL) models for medical imaging tasks have been shown to be prone to performance degradation on data from unfamiliar sources, such as from previously unseen institutions (Yu et al., 2022; Zech et al., 2018). Among the factors contributing to the reduction in performance are covariate and concept shifts (Cohen et al., 2020; Zhang et al., 2023); the former occurs due to shifts in the distribution of features between the training and testing datasets, while the latter occurs when the relationship between the input features and target variables changes. A substantial body of research has investigated the relationship between categorical attributes such as label distribution and patient demographics with performance and fairness gaps in the context of generalization (Pooch et al., 2020; Seyyed-Kalantari et al., 2020; Larrazabal et al., 2020; Ahluwalia et al., 2023; Seyyed-Kalantari et al., 2021; Glocker et al., 2023). Notably, shifts in these structured attributes

---

[*] Contributed equally

Code is available at: https://github.com/lotterlab/xray_generalization

across datasets often only account for a fraction of the generalization gaps, leaving room for understanding the scope of features that contribute to a lack of generalization (Wu et al., 2021). Traditionally, these generalization analyses are also often "model-centric", focusing primarily on assessing how the performance of one model varies across different datasets. While this approach offers insights into model robustness and adaptability, it may not fully capture the nuanced effects that the training domain has on the model behavior.

Here, we employ a "dataset-centric" approach to study how models trained on different domains behave when evaluated on the same dataset, helping to isolate the influence of training data. We specifically use two public chest x-ray datasets, CheXpert (Irvin et al., 2019) and MIMIC-CXR (Johnson et al., 2019), given their popularity and the high prevalence of commercial products for this modality. We first demonstrate that dataset-centric performance gaps exist for all pathologies in these datasets. Nonetheless, we find high variation in predictions at the image level independent of these gaps. We demonstrate that this variability is not simply noise, but can be predicted significantly above chance, where these "predictive tendencies" are specific to each pathology, cannot be explained by standard structured attributes alone, and seem to be encoded in high-level image characteristics.

## 2. Methods

### 2.1. Datasets

We used two public chest x-ray datasets: 1) CheXpert (CXP) (Irvin et al., 2019) with 224,314 images from 65,240 patients from Stanford Hospital patients and 2) MIMIC-CXR (MMC) (Johnson et al., 2019) consisting of 377,110 images from 65,379 patients at Beth Israel Deaconess Medical Center. Both datasets include metadata and labels for 14 radiological findings. When performing subgroup analysis, we considered structured attributes of radiographic projection (view), patient race, sex, and age. We included race subgroups of Asian, Black, and White patients in alignment with Gichoya et al. (2022) and 'AP', 'PA', and 'Lateral' views.

### 2.2. Deep Learning Model Training

We trained deep learning models for two distinct tasks as described below. These models are termed Pathology Prediction Models (PPMs) and Comparative Dataset Models (CDMs). Accordingly, we split each dataset into four subsets, consisting of 50% for training the PPMs, 20% for training the CDMs, 10% for validation for both model types, and 20% for testing for both model types. These partitions were performed on a patient-level to ensure that all radiographs from one patient were included in the same partition.

**Pathology Prediction Models (PPMs)** The PPM models are trained according to the standard task of predicting the presence of pathologies. We train these models using the following 12 pathology labels: 'Atelectasis', 'Cardiomegaly', 'Consolidation', 'Edema', 'Enlarged Cardiomediastinum', 'Fracture', Lung Lesion', 'Lung Opacity', 'Pleural Effusion', 'Pleural Other', 'Pneumonia', 'Pneumothorax'. For robustness, we mapped uncertain labels (-1) as missing. We trained models separately on CXP and MMC and refer to them as CXP-PPM and MMC-PPM.

**Comparative Dataset Models (CDMs)**   The CDMs are designed to discern whether, for a given pathology, an x-ray is more likely to yield a relatively higher prediction from the CXP-PPM or the MMC-PPM. We refer to this difference in the PPM predictions as the *predictive tendency.* To generate the CDM training labels for a given dataset (Figure 1), we first rank the PPM predictions for the CXP- and MMC-PPMs for a given pathology $p$, with the normalized rankings denoted as $\Phi^p$. This step helps to mitigate calibration discrepancies between the two PPMs. Subsequently, we compute the *predictive tendency* $s_p(x_i)$ for image $x_i$ and pathology $p$ as follows:

$$s_p(x_i) = \Phi^p_{CXP}(x_i) - \Phi^p_{MMC}(x_i) \tag{1}$$

The resulting values are binarized into discrete labels, denoted as $s'_p(x_i)$, using the median value from the CDM-training dataset as the binarization threshold, which results in a balanced distribution of labels. Each CDM was trained on a pathology-specific subset of the full dataset. For example, for pneumothorax, we compute the predictive tendency labels for all images with pneumothorax and use these labels to train a pneumothorax-specific CDM. The trained CDM outputs an inferred predictive tendency $\hat{s}_p'(x_i)$ to estimate whether a model trained on CXP or MMC outputs a relatively higher pathology prediction for that image.

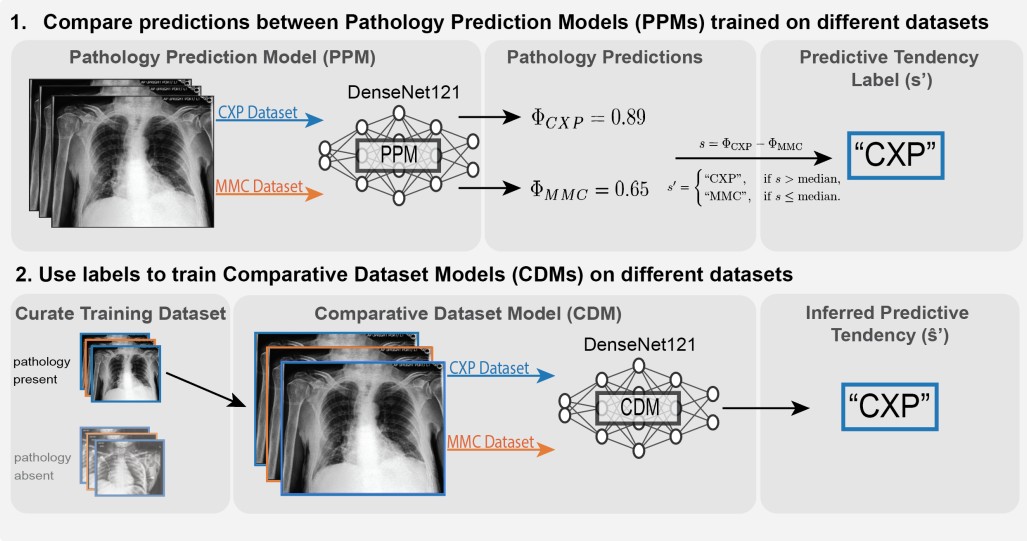

Figure 1: **Development of Pathology Prediction Models (PPMs) and Comparative Dataset Models (CDMs).** PPMs are trained to predict the presence of pathologies. The differences in prediction scores between PPMs trained on different datasets are used to derive labels, which are then used to train the CDMs.

**Model Architecture and Training**   For both training tasks (PPM and CDM), we employed DenseNet121 models (Huang et al., 2016), pre-trained on ImageNet. Models were trained for 50 epochs and training was performed separately on each dataset. Final weights were selected based on the highest performance, as determined by AUROC, on the validation dataset. For each model configuration (PPM/CDM and CXP/MMC), we train three

identical models using different random seeds to increase the robustness of the analysis. Unless specified otherwise, reported performance metrics represent the average across the three models. When using the PPMs to create training labels for the CDMs, the PPM predictions are averaged across the three random seeds and subsequently ranked. Models are implemented and trained using the TorchXrayVision library (Cohen et al., 2021). Overall, our goal was to develop models representing standard architectures, hyperparameters, and image preprocessing, where more information can be found in the Appendix A.

**Image Transformations** To study the importance of image characteristics for a CDM's ability to learn predictive tendencies, we applied the following image transformations: 1) Pixel permutations: to disrupt the spatial information without affecting intensity distributions by randomizing the position of each pixel within an image; 2) Frequency filtering: to selectively remove low and high-frequency content (see Appendix A for more details).

### 2.3. Statistical Analysis

To test the association between the distribution of predictive tendencies and categorical structured attributes (view, race, sex), we first performed a Kruskal-Wallis test (Kruskal and Wallis, 1952), followed by a post-hoc Dunn test (Dunn, 1964). We use $\epsilon^2$ (King and Minium, 1981) to determine effect sizes. Associations of the predictive tendency with patient age are tested using the Spearman correlation coefficient. We apply Bonferroni correction to correct for multiple comparisons. Statistical analysis was conducted using python 3.9 with SciPy 1.10 and scikit-posthocs 0.8.1 (Virtanen et al., 2020).

## 3. Results

### 3.1. Pathology Prediction Model performance and performance gaps

The PPMs trained on CXP and MMC achieved overall AUROC scores of 0.89 and 0.93 respectively (micro-average across all 12 pathologies), with AUROCs for each pathology detailed in Appendix B, Table 2. The performances of the PPMs are comparable to those reported in the existing literature, despite using only 50% of the available data for training (Seyyed-Kalantari et al., 2020; Pham et al., 2019).

The generalization gap for a model $f$ and performance metric $R$ is traditionally defined by the difference: $R(f_A, D_A) - R(f_A, D_B)$, where $D_A$ and $D_B$ are distinct, mutually exclusive domains, and $f_A$ denotes the model developed on $D_A$. We focus on a dataset-centric approach for comparing model performance: $R(f_A, D_A) - R(f_B, D_A)$, where the models are trained separately on $D_A$ and $D_B$ and evaluated on the same domain. This dataset-centric perspective shifts the focus to the training datasets to offer a more direct comparison of models on the same data points.

The performance gaps between in-domain and out-of-domain models across all 12 pathologies are depicted in Figure 2A. In-domain models consistently outperformed out-of-domain ones, though the extent of these gaps varied by pathology. Fracture and pneumothorax exhibited the largest gaps, while effusion and atelectasis had the smallest. Notably, pathologies with higher PPM model performance tended to show narrower gaps (Figure 2B).

### 3.2. Predictive Tendencies

Beyond aggregate performance differences, we studied the model behavior at the individual image level. In this analysis, we focused on three pathologies due to their varying performance gaps: pneumothorax, pneumonia, and pleural effusion. We first visualized the image-level prediction scores between the different models. As illustrated in Figure 2C and D, notable image-level variations were observed, even for pleural effusion which exhibits high predictive performance and a relatively smaller performance gap. To quantify these differences, we introduce the concept of *predictive tendency* ($s$) (Section 2.2) that measures the difference in the ranked prediction scores between PPMs trained on different datasets. In the analysis that follows, we analyze these predictive tendencies for images that are labeled positive for each of the three selected pathologies.

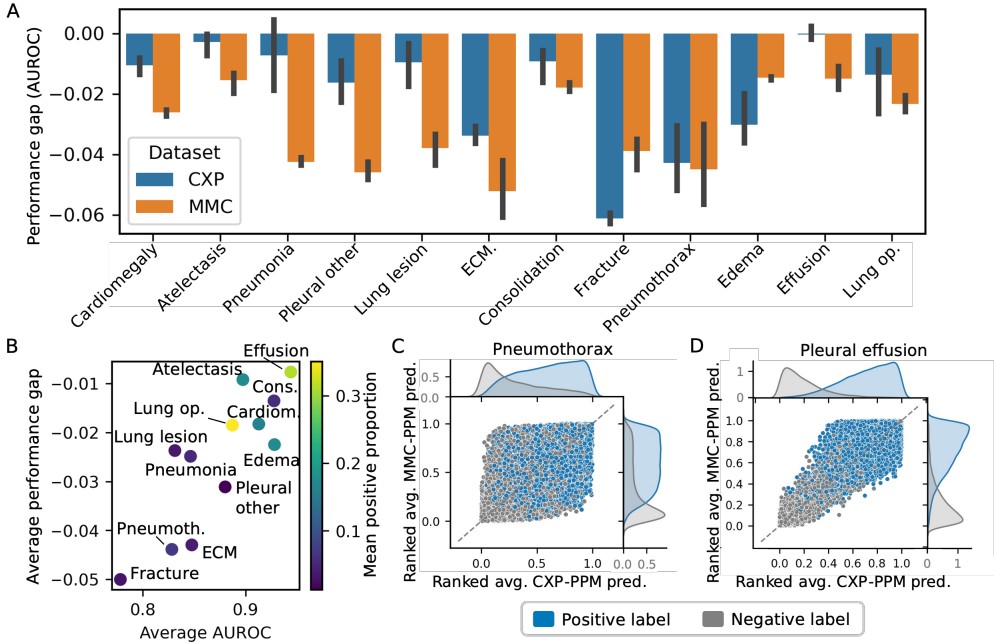

Figure 2: **Performance Gaps.** A) Comparison of CXP- and MMC-PPM performance. Each bar represents the difference in performance between the out-of-domain model minus the in-domain model on the CXP (blue) and MMC (orange) test datasets. B) Association between pathology AUROC and performance gap (averaged across CXP- and MMC-PPMs). Color indicates the relative amount of positive labels in the datasets (# of positive labels/total # of images, a proxy for amount of training data). C,D) Ranked CXP- and MMC-PPM predictions for CXP pneumothorax and pleural effusion for positive (blue) and negative (grey) images. Cardiom: Cardiomegaly, Cons: Consolidation, ECM: Enlarged cardiomediastinum, Lung op: Lung opacity, Pneumoth: Pneumothorax

**Structured Attributes**   To examine whether the observed predictive tendencies are random or reflect systematic features, we first explored whether the tendencies are correlated with structured attributes: patient demographics (race, sex, age) and x-ray radiographic view. Akin to prior analysis of generalization gaps with these datasets, we find that these

attributes can only explain a small fraction of the observed predictive tendencies. Across the categorical attributes (race, sex, view), the attribute generally showing the highest association is view (Figure 3 and Appendix D, Figure 9). Nonetheless, this effect was really only visually apparent for pleural effusion with an $\epsilon^2$ effect size of 0.18 in MMC (with 0 being chance and 1 corresponding to all variance explained). The next highest effect size was 0.06 for patient sex in the CXP dataset for pneumothorax, with generally low visually apparent differences in the distribution of predictive tendencies (Appendix D, Figure 9). All other $\epsilon^2$ effect sizes across patient sex and race were below 0.01. Correlations with patient age were also generally low, with an Spearman correlation coefficients ranging from 0.02-0.21 across datasets and pathologies (Appendix D, Table 3 and Fig. 10).

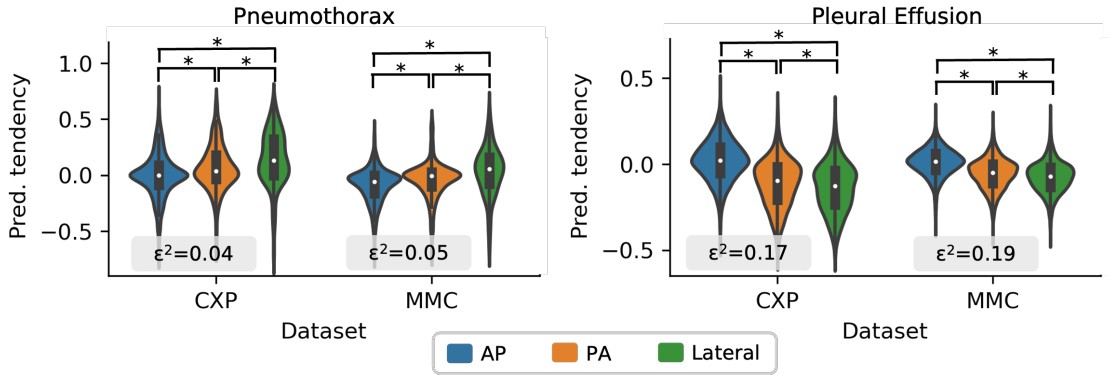

Figure 3: **Predictive Tendencies by Radiographic View.** Distributions of predictive tendencies for each radiographic view for images with pneumothorax (left) and pleural effusion (right). *: p < 0.02 (Kruskal-Wallis test followed by pairwise Dunn test), effect sizes determined by $\epsilon^2$.

**Intensity Distributions** As the structured attributes only explain a small fraction of the observed variance, we next investigated a possible correlation between the predictive tendencies and first-order image statistics, namely the distribution of pixel intensities. If there were such a correlation, one would expect differences in the intensity histograms between images that have a predictive tendency towards MMC versus towards CXP. While we do observe differences in the histograms between the CXP and MMC datasets themselves, the variation between images with different predictive tendencies was negligible compared to these disparities between the datasets, as shown in Appendix E. Notably, the differences in intensity histograms between datasets were consistently larger than the differences between predictive tendencies even after adjusting for structured attributes.

### 3.3. Comparative Dataset Models

As neither structured attributes nor first-order image statistics could explain the predictive tendencies, we next asked whether they are determined by patterns that DL models could recognize. To this end, we trained a separate set of deep learning models termed Comparative Dataset Models (CDMs) (Figure 1 and Section 2.2). CDMs were trained separately for each dataset and for each of the three selected pathologies to avoid introducing the dataset source as a potential shortcut and to enable comparisons across pathologies.

Surprisingly, we find that the predictive tendencies can indeed be predicted well above chance levels, with AUROCs between 0.74 and 0.85 (pneumonia and pleural effusion MMC-CDMs, see Table 1). Notably, CDMs trained on CXP or MMC showed meaningful cross-dataset performance, suggesting generalizable predictive patterns (grey values in Table 1).

Table 1: **Performance of Comparative Dataset Models.** AUROCs for CDMs for pathologies evaluated on both datasets.

| | Pneumothorax evaluated on | | Pneumonia evaluated on | | Pleural Effusion evaluated on | |
|---|---|---|---|---|---|---|
| **CDM Training Data** | CXP | MMC | CXP | MMC | CXP | MMC |
| CXP | 0.841 | 0.695 | 0.751 | 0.626 | 0.841 | 0.792 |
| MMC | 0.675 | 0.799 | 0.725 | 0.749 | 0.789 | 0.845 |

With this highly non-trivial CDM performance, we next explored what types of information these models have learned. We first examined exemplary images, comparing those that received high predictions from the CDMs against those with low predictions. This was followed by examining the associations between CDM predictions and structured attributes. These analyses aligned with our previous findings regarding the predictive tendencies, where radiographic view had a moderate association with CDM predictions but other associations were much less apparent. More details and visual representations can be found in Appendices F and G.

Next, we assessed whether the models generalize across pathologies, i.e., if a CDM model trained on images with pneumothorax could also predict the predictive tendencies for images with pleural effusion. Interestingly, we find that this is not the case. As shown in Appendix H, Tables 5 and 6, the models exhibit nearly random AUROC performance with a range of 0.43-0.60. These results further suggest that the predictive tendencies are pathology-specific and cannot be explained by low level statistical differences.

Subsequently, we tested how other image characteristics might influence CDMs' ability to discern predictive tendencies by applying targeted image transformations, retraining the CDMs on these transformed images and assessing the impact on CDM performance. To examine the role of spatial relationships, we randomized pixel positions in the x-rays, disrupting the spatial structure but preserving the intensity distribution. This pixel permutation significantly reduced CDM performance, dropping AUROCs to near-chance levels, between 0.55 and 0.64 (Appendix H, Table 7), again aligning with our previous findings that predictive tendencies are not linked to low-level image statistics.

We then assessed the importance of image frequencies by filtering out high or low frequencies, following the method of Gichoya et al. (2022) who used this explainability approach in the context of models trained to predict patient race. When applied to the CDMs, the filtering variably affected performance but consistently showed that CDMs rely on both frequency ranges (Figure 4). Notably, CDMs maintained non-trivial performance even when features are largely not perceptible to the human eye, qualitatively similar to the results of Gichoya et al. (2022) for predicting patient race and suggesting that both sets of models may rely on a mix of statistical features for their respective tasks.

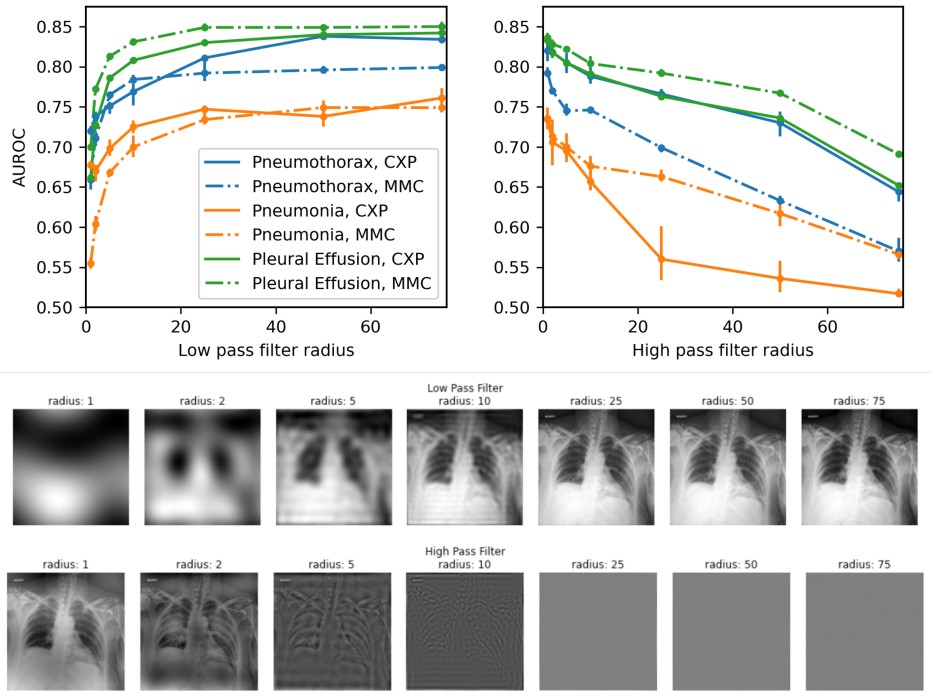

Figure 4: **Performance of CDMs Trained and Tested Using Modified Images** Top: AUROCs with varying filter radius (low (left) and high pass (right) filters). Bottom: Sample images.

## 4. Discussion

Despite a wealth of research into DL model performance gaps, particularly in chest x-ray analysis, a significant portion of these gaps remain unexplained (Wu et al., 2021). In this study, we adopt a "dataset-centric" perspective to examine performance gaps between DL models trained on different datasets, revealing that these gaps vary across pathologies and are influenced by their representation in the datasets. Beyond performance however, a key contribution of our work is introducing the concept of "predictive tendency" – the notion that models trained on different datasets can exhibit systematic differences in predictions regardless of overall performance. Our analysis suggests that these tendencies in two popular datasets are pathology-specific and depend on higher-order image statistics that are not fully captured by structured attributes. These results especially emphasize the heterogeneity of disease presentation (Oakden-Rayner et al., 2019) and label creation, where variations in representation across datasets and even radiologist/clinic-specific interpretation practice could potentially contribute to our findings. Beyond showing that the tendencies can be predicted using separate deep learning models, we envision avenues where this approach can enable context-dependent ensembling strategies of multiple models, and tailored model selection to individual patient and image profiles more generally. Altogether, our analysis highlights the importance of a nuanced view of generalization, emphasizing the need to address context-specific biases while leveraging this knowledge increase performance robustness across all patients.

## Acknowledgments

We thank Christopher Bridge for his thoughtful feedback.

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

## Appendix A. Model training details

**Image preprocessing**   Following common practices in the field (Cohen et al., 2021), image preprocessing consisted of the following steps in order:

1. Center cropping to a square image

2. Resizing to 224 by 224 pixels

3. Intensity normalization to a range of -1024 to 1024

**Hyperparameters**   We used the following hyperparameters for model training:

- Loss function: Weighted categorical cross-entropy; weights were determined by the inverse of the label frequency in the training dataset

- Adam optimizer

- Learning rate: 1.0e-3

- Weight decay: 1.0e-5

- Number of epochs: 50

**High and low pass filters**   We first convert an image to the frequency domain. Subsequently, we defined a circular filter of a specified radius, $r$, and removed the frequency content within the circle (high-pass) or outside of it (low-pass filtering). Lastly, we applied an inverse Fourier transformation to convert the image back to the spatial domain. For this step we utilize code provided by Gichoya et al. (2022).

## Appendix B. PPM results

Table 2: Performance of PPMs Per Dataset and Pathology: AUROCs (averaged across seeds) per CXP- and MMC-PPMs tested on both datasets.

| Pathology | CXP test dataset | | MMC test dataset | |
|---|---|---|---|---|
| | CXP-PPM | MMC-PPM | CXP-PPM | MMC-PPM |
| Atelectasis | 0.887 | 0.884 | 0.890 | 0.908 |
| Cardiomegaly | 0.925 | 0.915 | 0.874 | 0.900 |
| Consolidation | 0.921 | 0.915 | 0.918 | 0.934 |
| Edema | 0.915 | 0.891 | 0.925 | 0.941 |
| Enlarged Cardiomediastinum | 0.815 | 0.782 | 0.824 | 0.881 |
| Fracture | 0.795 | 0.759 | 0.738 | 0.761 |
| Lung Lesion | 0.822 | 0.819 | 0.799 | 0.841 |
| Lung Opacity | 0.895 | 0.879 | 0.856 | 0.880 |
| Pleural Effusion | 0.936 | 0.937 | 0.935 | 0.952 |
| Pleural Other | 0.859 | 0.831 | 0.856 | 0.901 |
| Pneumonia | 0.861 | 0.858 | 0.792 | 0.832 |
| Pneumothorax | 0.790 | 0.755 | 0.814 | 0.866 |

## Appendix C. PPM prediction distributions

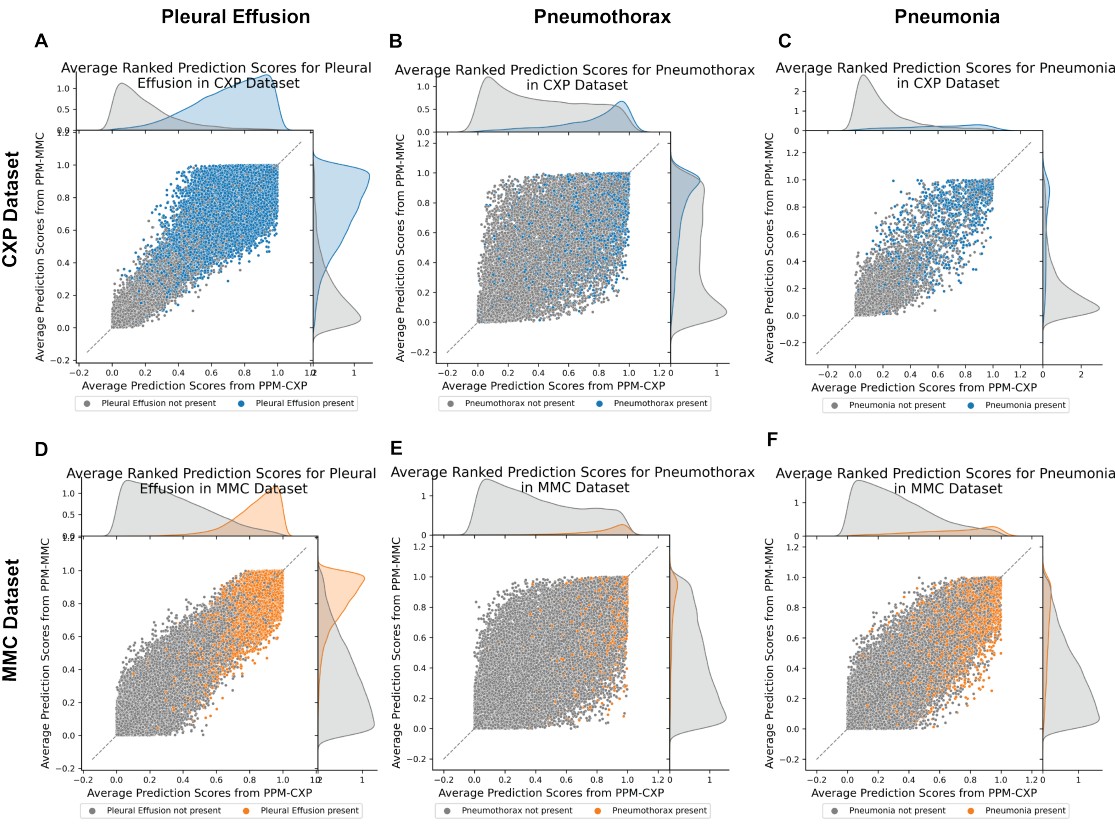

Figure 5: Ranked Predictive Scores Across Pathologies and Dataset: (A) Pleural Effusion in CXP Dataset, positive (blue), negative (grey). (B) Pneumothorax in CXP Dataset, positive (blue), negative (grey). (C) Pneumonia in CXP Dataset, positive (blue), negative (grey). (D) Pleural Effusion in MMC Dataset, positive (orange), negative (grey). (E) Pneumothorax in MMC Dataset, positive (orange), negative (grey). (F) Pneumonia in MMC Dataset, positive (orange), negative (grey). effusion (D) positive (blue) and negative (grey) images

## Appendix D. Structured Attributes

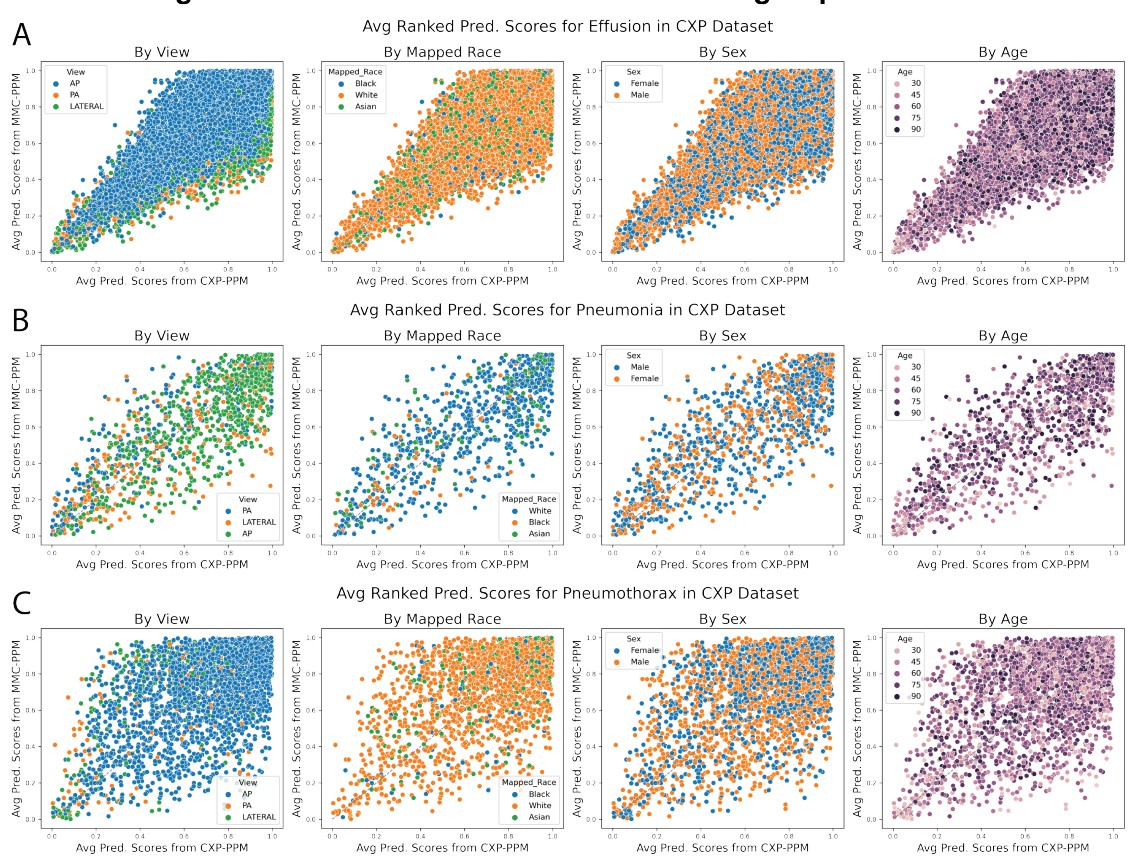

Figure 6: Subgroup Distributions Across Dataset: (A) Sex, (B) Race , (C) Age , (D) View.

Figure 7: Ranked Predictive Scores for subgroups in CXP dataset: (A) Pleural Effusion By View, Race, Sex and Age, (B) Pneumonia By View, Race, Sex and Age, (C) Pneumothorax By View, Race, Sex and Age.

**Average Ranked Prediction Scores Across Subgroup in MMC Dataset**

Figure 8: Ranked Predictive Scores for subgroups in MMC dataset: (A) Pleural Effusion By View, Race, Sex and Age, (B) Pneumonia By View, Race, Sex and Age, (C) Pneumothorax By View, Race, Sex and Age.

Table 3: Spearman correlation coefficients between age and predictive tendencies $s$ for images with pneumothorax, pneumonia, and pleural effusion.

| Dataset | Pneumothorax | Pneumonia | Pleural Effusion |
|---------|--------------|-----------|------------------|
| CXP     | 0.05         | 0.21      | -0.02            |
| MMC     | 0.14         | 0.15      | -0.07            |

Figure 9: Predictive Tendency Across Subgroups: (A) Pleural effusion by view, race, and sex, (B) Pneumonia by view, race, and sex, (C) Pneumothorax by view, race, and sex. Epsilon squared effect sizes are displayed when associations between the predictive tendency and structured attributes are statistically significant.

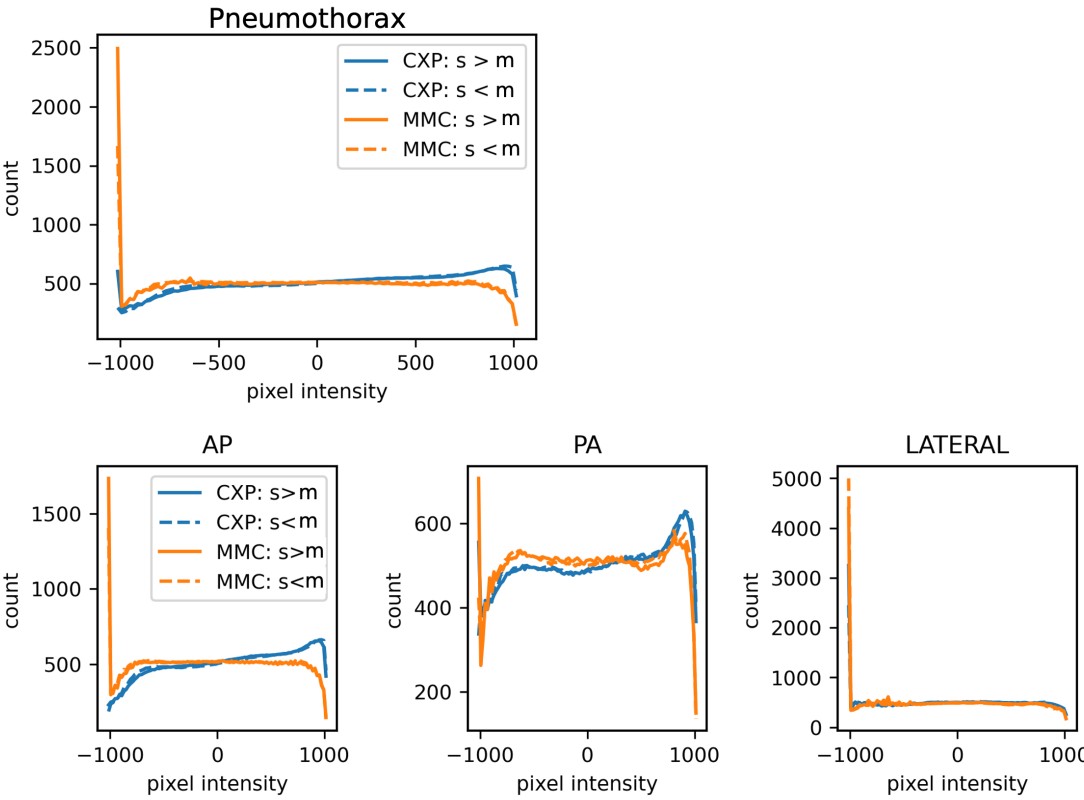

Figure 10: Relationship between the predictive tendency and patient age for images with pneumothorax (top), pneumonia (middle), and pleural effusion (bottom) for test data from the CXP (left) and MMC dataset (right).

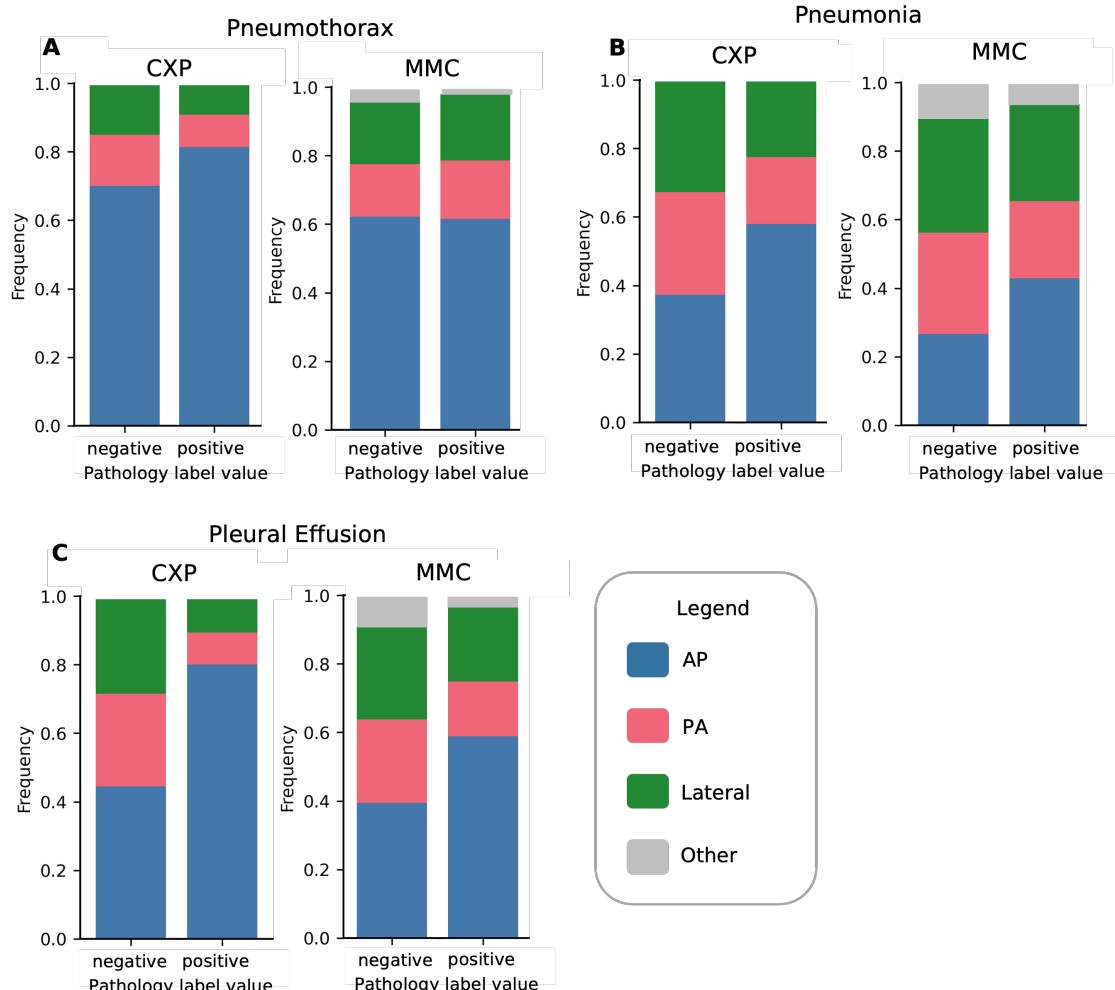

Figure 11: View distribution differences between images with pathology negative and positive images for pneumothorax (A), pneumonia (B), and pleural effusion (C).

## Appendix E. Histograms

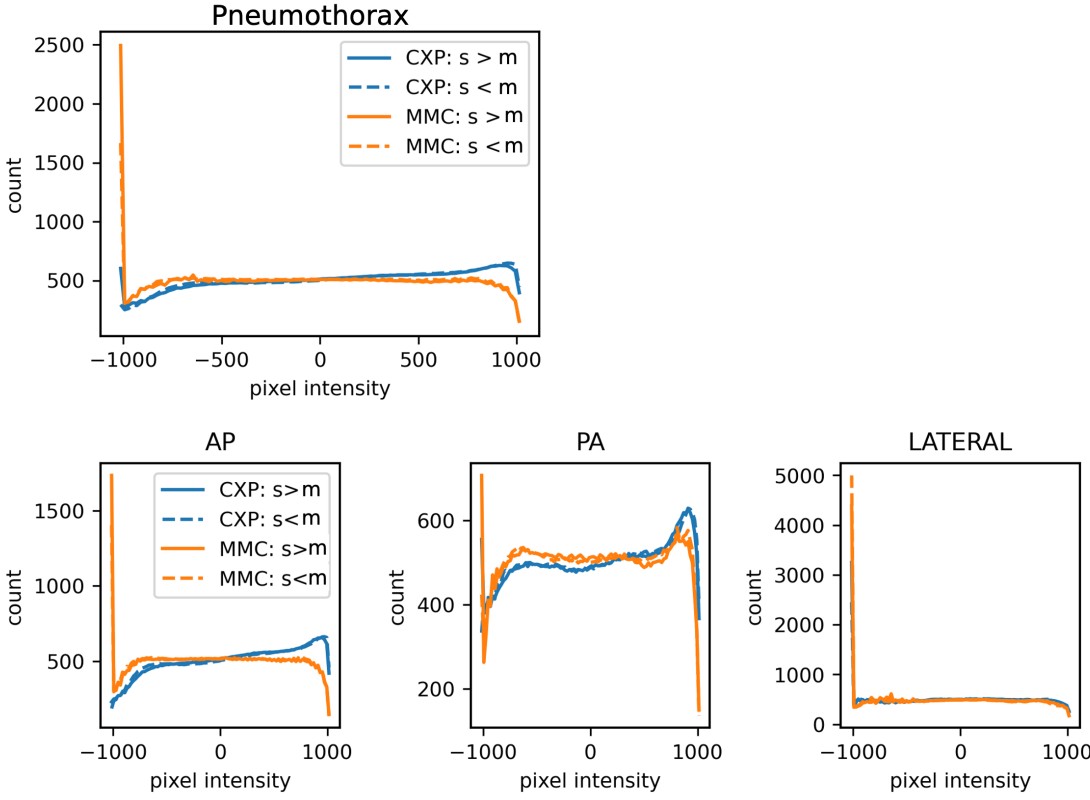

Figure 12: Average histograms for pneumothorax positive images by predictive tendency (top) and separate for each view (bottom row) for the CXP (blue) and MMC test datasets (orange). m denotes the binarization threshold of the predictive tendency s (median predictive tendency).

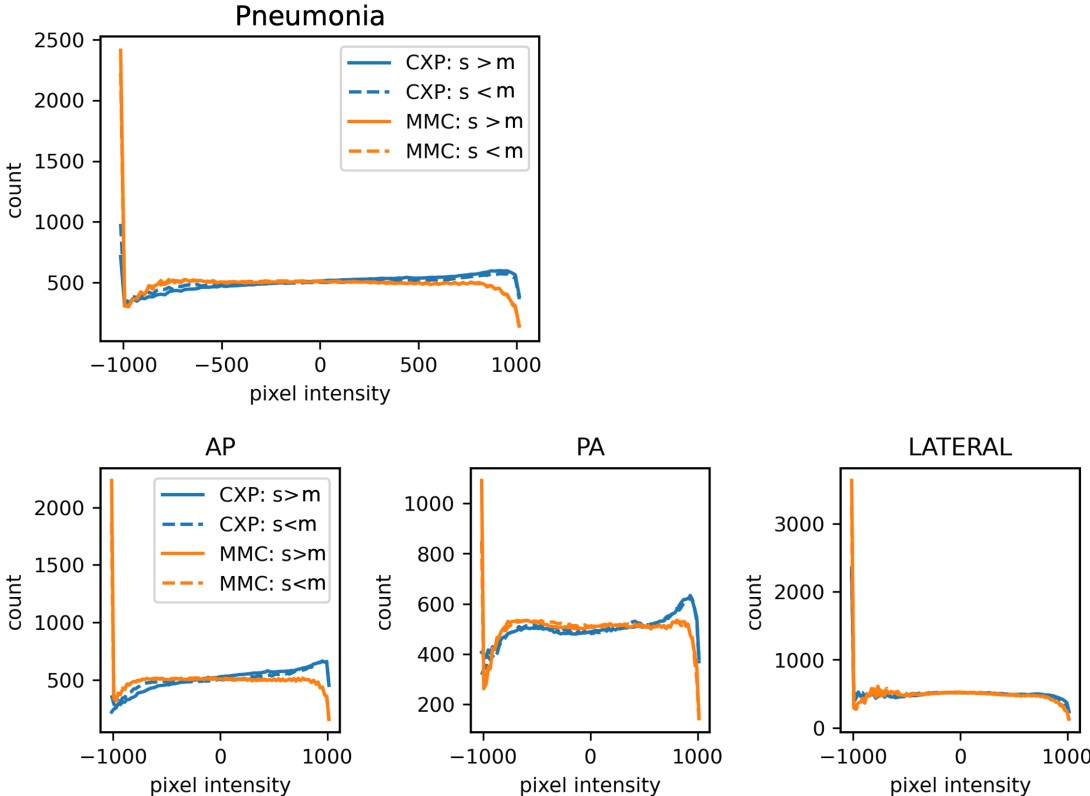

Figure 13: Average histograms for pneumonia positive images by predictive tendency (top) and separate for each view (bottom row) for the CXP (blue) and MMC test datasets (orange). m denotes the binarization threshold of the predictive tendency s (median predictive tendency)..

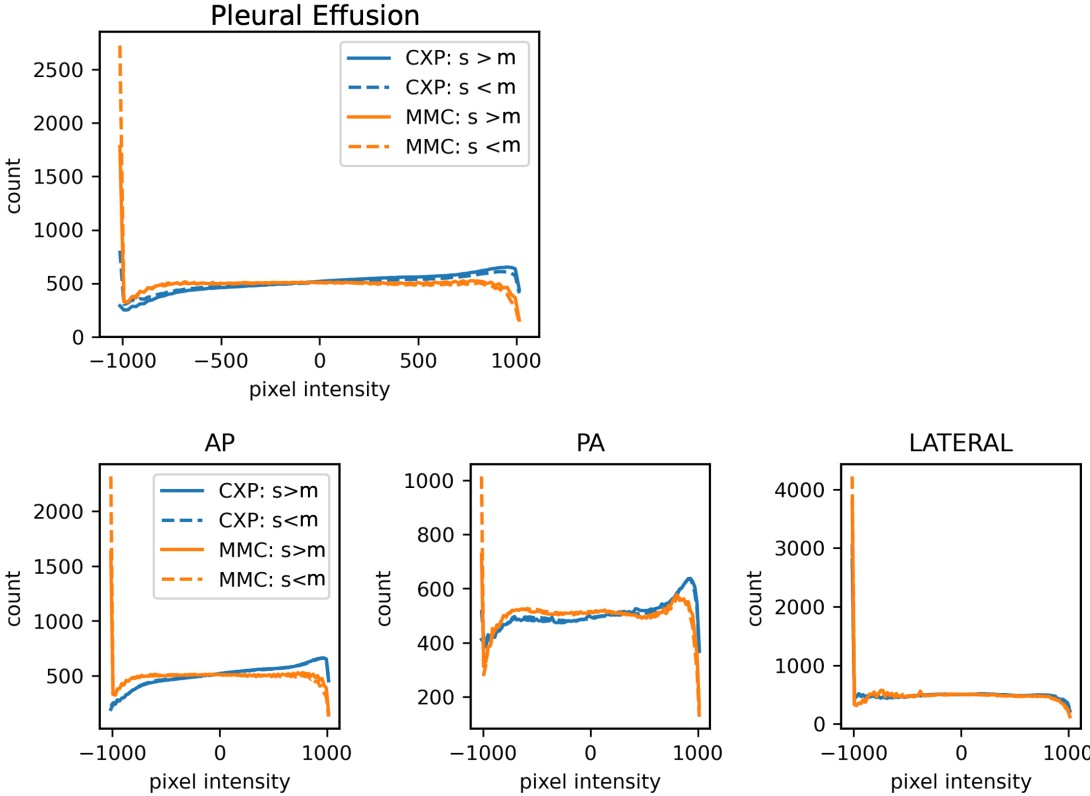

Figure 14: Average histograms for pleural effusion positive images by predictive tendency (top) and separate for each view (bottom row) for the CXP (blue) and MMC test datasets (orange). m denotes the binarization threshold of the predictive tendency s (median predictive tendency)..

## Appendix F. Comparison of X-rays with high and low CDM prediction values

Example images that received high and low CDM predictions are plotted below, where high predictions correspond to an inferred CXP predictive tendency and low predictions correspond to an inferred MMC predictive tendency (Figures 15, 16, 17, and 18). While there is large heterogeneity amongst these images, there are several contexts where particular radiographic views are overrepresented. For instance, images depicting pneumothorax that were inferred to have a CXP predictive tendency were predominantly lateral views. Conversely, pneumonia and pleural effusion images that were inferred to have a MMC predictive tendencywere more likely to be lateral views. Nonetheless, the features driving other CDM predictions are often unclear.

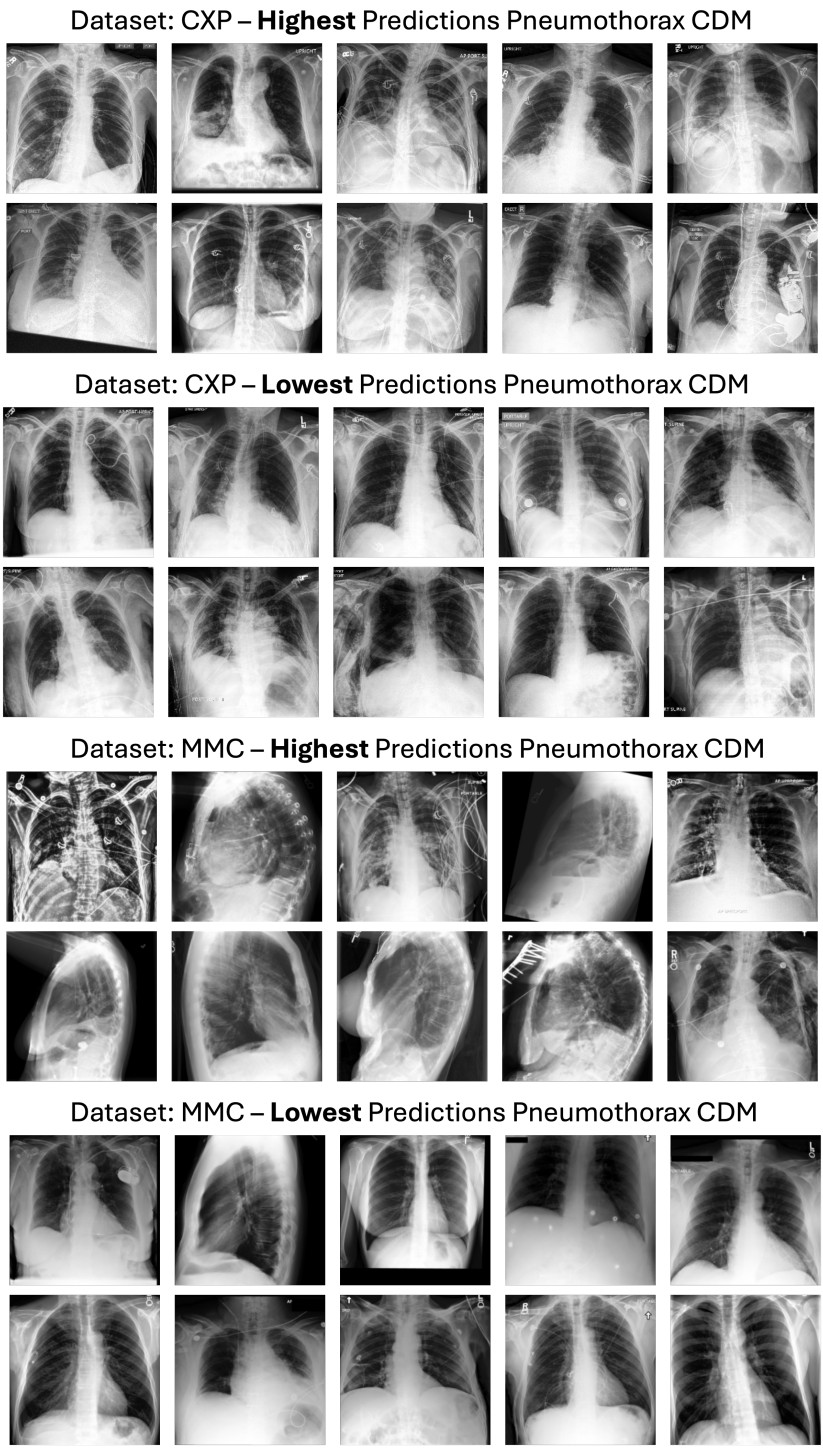

Figure 15: **Examples for the Pneumothorax CDM** X-rays were randomly selected among the 10% of test dataset images with the highest (lowest) predictions of the Comparative Dataset Model trained to infer the predictive tendency of images with pneumothorax.

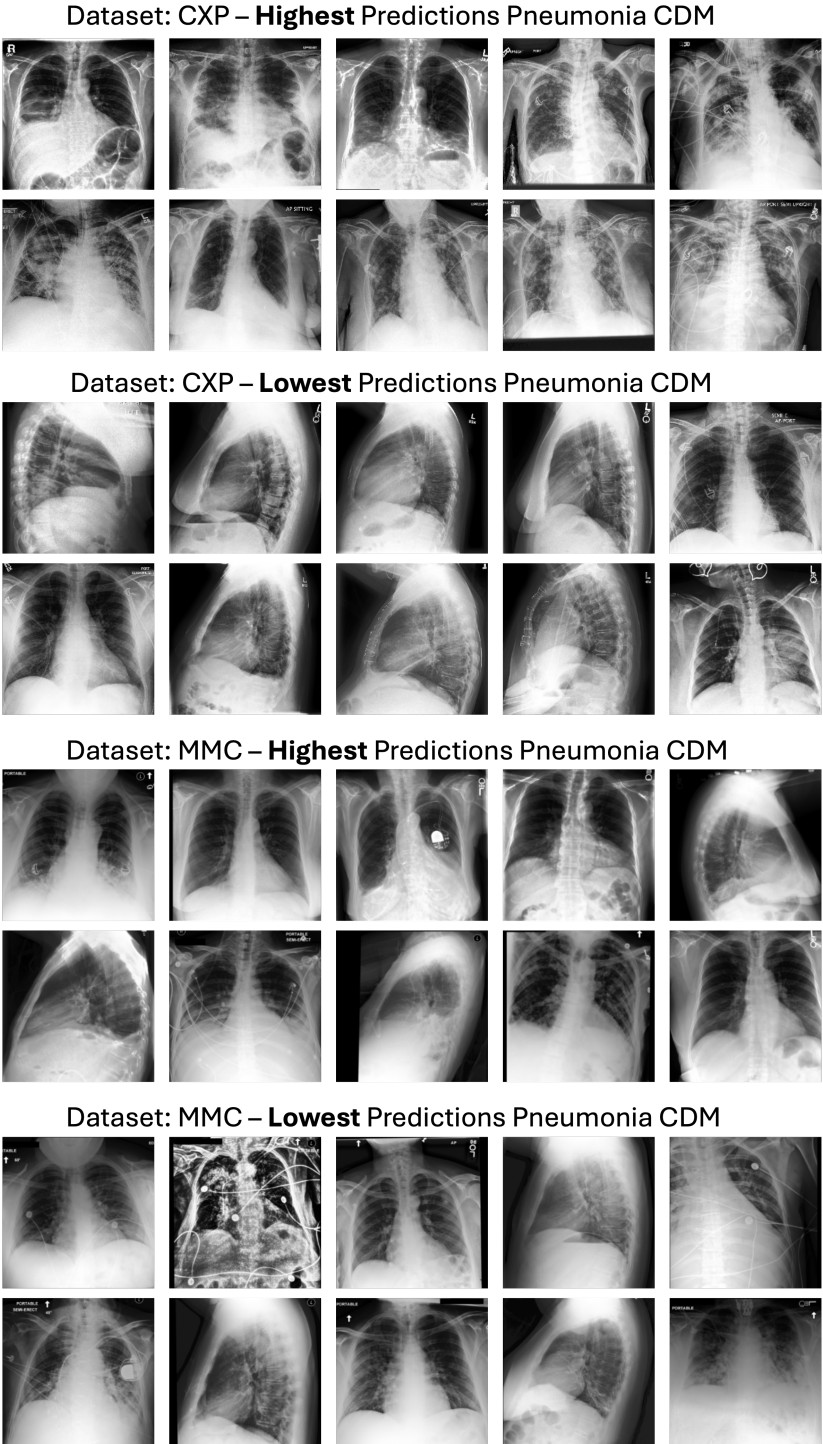

Figure 16: **Examples for the Pneumonia CDM** X-rays were randomly selected among the 10% of test dataset images with the highest (lowest) predictions of the Comparative Dataset Model trained to infer the predictive tendency of images with pneumonia.

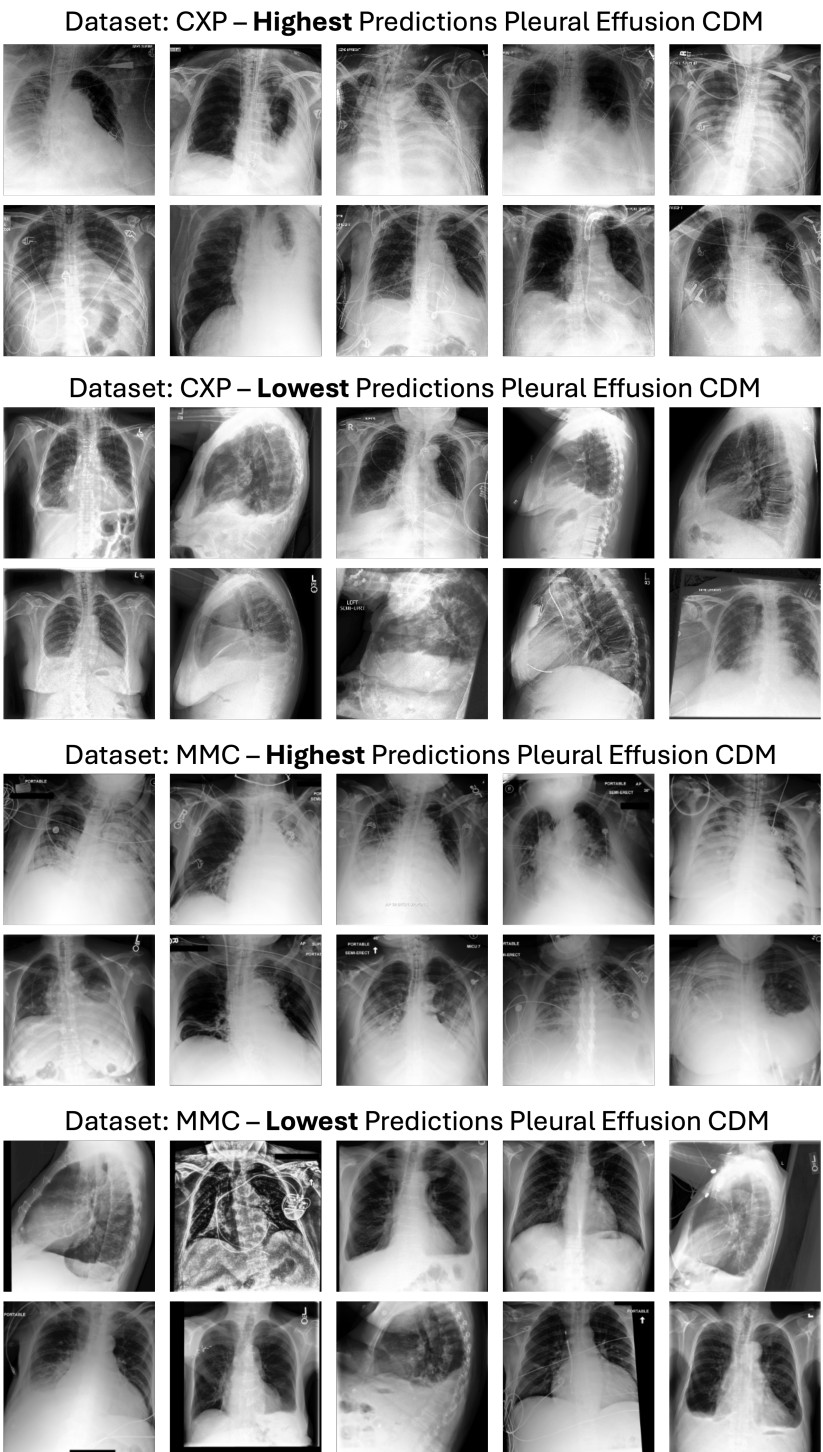

Figure 17: **Examples for the Pleural Effusion CDM** X-rays were randomly selected among the 10% of test dataset images with the highest (lowest) predictions of the Comparative Dataset Model trained to infer the predictive tendency of images with pleural effusion

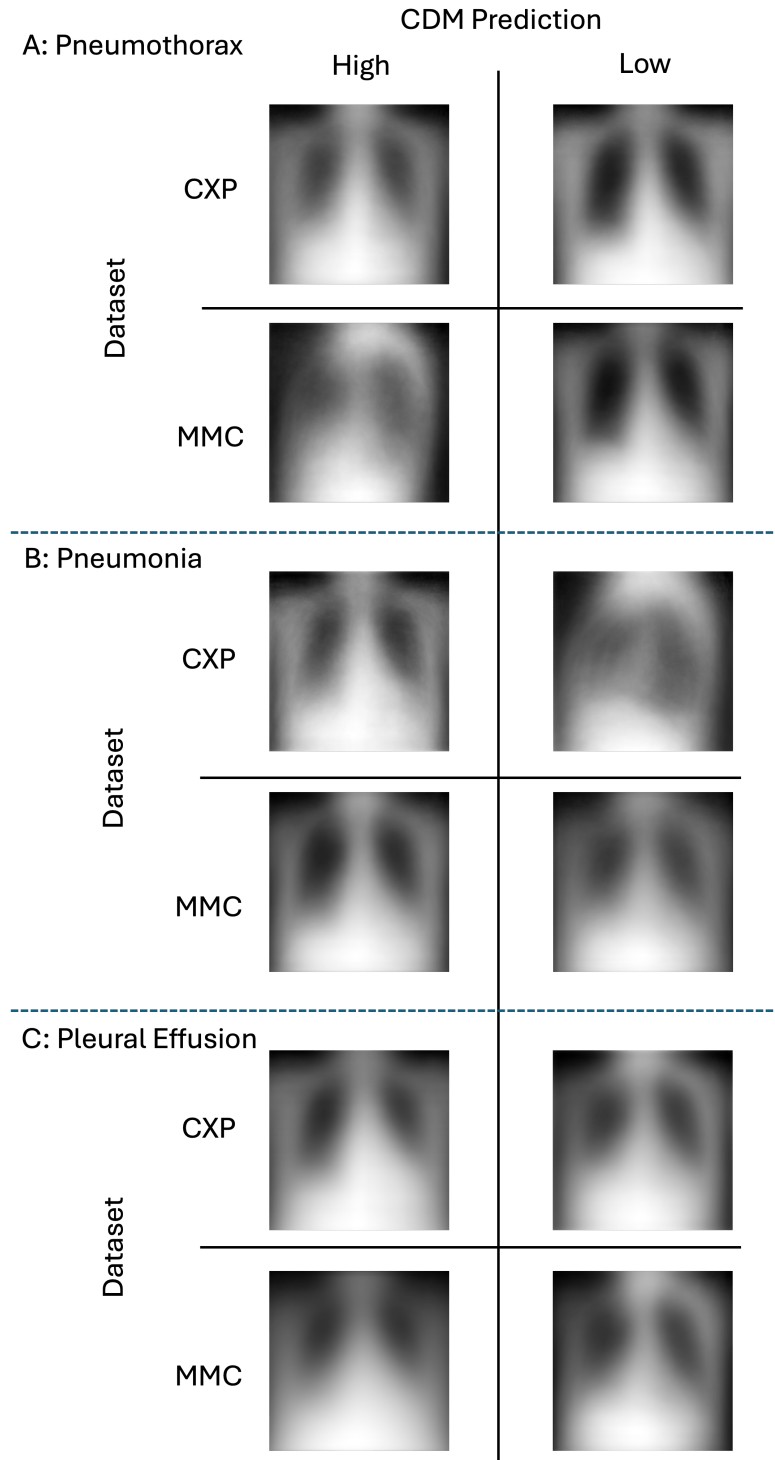

Figure 18: **Average of the X-rays with highest/lowest CDM predictions** Averages of the 10% of test dataset images with the highest (lowest) predictions for CDMs trained on images with pneumothorax (A), pneumonia (B), and pleural effusion (C).

## Appendix G. Association between CDM predictions and structured attributes

Similarly to the analysis of associations between the predictive tendency and structured attributes in Section 3.2, we analyzed the associations between CDM predictions (inferred predictive tendencies) and structured attributes. These analyses largely reflected the effects outlined in Section 3.2 regarding predictive tendencies, wherein structured attributes explained only a minor portion of the CDM predictions. Among the attributes examined, the radiographic view emerged as having the strongest association with CDM predictions across all pathologies. However, the effect sizes, quantified by epsilon squared, were modest, ranging between 0.07 and 0.32. The patient's sex showed a statistically significant association with the CDM predictions for pneumonia and pleural effusion, albeit with minimal effect sizes (epsilon squared of 0.006 and 0.02, respectively). No statistically significant associations were observed between CDM predictions and the patient's race or age for any pathology, further emphasizing the unique association of the radiographic view.

Table 4: Spearman correlation coefficients between age and predictive tendencies $s$ for images with pneumothorax, pneumonia, and pleural effusion.

| Dataset | Pneumothorax | Pneumonia | Pleural Effusion |
|---------|--------------|-----------|------------------|
| CXP | 0.06 | 0.22 | -0.03 |
| MMC | 0.15 | 0.20 | -0.06 |

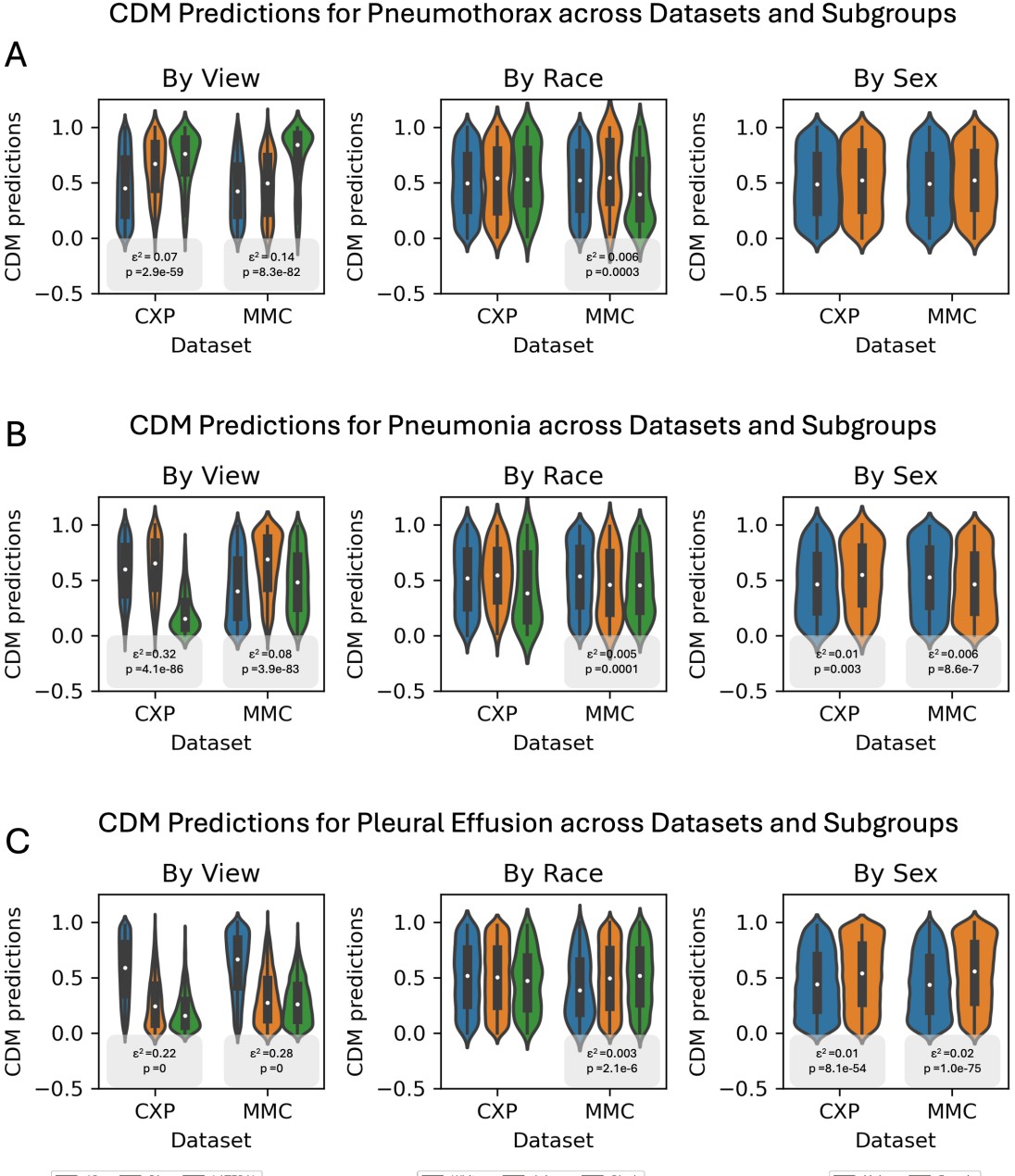

Figure 19: Comparative Dataset Model Predictions Across Subgroups: (A) Images with pleural effusion by view, race, and sex, (B) image with pneumonia by view, race, and sex, (C) images with pneumothorax by view, race, and sex. Epsilon squared effect sizes are displayed when associations between the CDM prediction (inferred predictive tendency) and structured attributes are statistically significant.

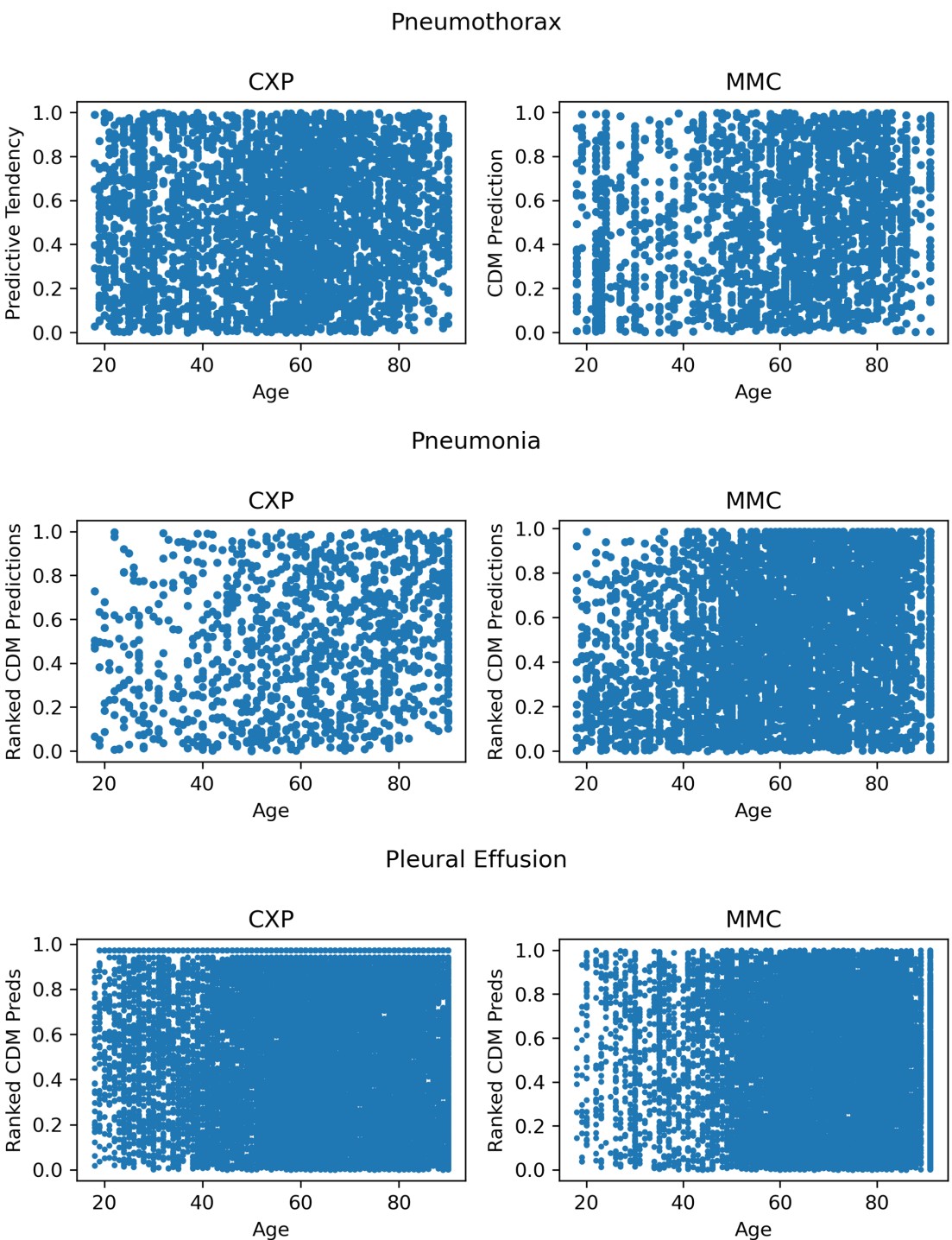

Figure 20: Relationship between CDM predictions and patient age for images with pneumothorax (top), pneumonia (middle), and pleural effusion (bottom) for test data from the CXP (left) and MMC dataset (right).

## Appendix H. Additional CDM results

Table 5: Performance of CXP-CDM Generalization Across Pathologies in CXP Dataset.

| Models | Datasets | | |
|---|---|---|---|
| | Pneumothorax | Pneumonia | Pleural Effusion |
| **Pneumothorax** | N/A | 0.601 | 0.432 |
| **Pneumonia** | 0.550 | N/A | 0.539 |
| **Pleural Effusion** | 0.495 | 0.560 | N/A |

Table 6: Performance of MMC-CDM Generalization Across Pathologies in MMC Dataset.

| Models | Datasets | | |
|---|---|---|---|
| | Pneumothorax | Pneumonia | Pleural Effusion |
| **Pneumothorax** | N/A | 0.537 | 0.369 |
| **Pneumonia** | 0.584 | N/A | 0.479 |
| **Pleural Effusion** | 0.473 | 0.524 | N/A |

Table 7: Performance of CDMs with Pixel Permutations.

| Model trained on* | Pneumothorax evaluated on | | Pneumonia evaluated on | | Pleural Effusion evaluated on | |
|---|---|---|---|---|---|---|
| | CXP | MMC | CXP | MMC | CXP | MMC |
| CXP | 0.627 | 0.545 | 0.625 | 0.539 | 0.644 | 0.609 |
| MMC | 0.561 | 0.554 | 0.561 | 0.548 | 0.640 | 0.632 |

*Each row lists the performances of CDMs trained on either CXP or MMC subsets. For example, the pneumothorax CXP-CDM was trained and evaluated on pneumothorax positive images form the CDM-training and test datasets. Gray values indicate out-of-domain performance, i.e. the pneumothroax CXP-CDMs tested on MMC pneumothorax test data and vice versa.

