# OpenReview forum: "Beyond Structured Attributes: Image-Based Predictive Trends for Chest X-Ray Classification"
_MIDL.io/2024/Conference — MIDL 2024 Oral_

### Official Review · Reviewer_ogFB · 2024-02-26

**Confidence:** 4
**Preliminary Rating:** 4
**Recommendation:** Oral

**Summary:**

This study sheds light on systematic differences observed in models trained on different datasets. The investigation focuses on chest X-ray classification using two popular datasets: CheXpert and MIMIC-CXR. The findings reveal that variations at the image level are not arbitrary but are correlated with specific pathologies, indicating consistent tendencies in models trained on distinct datasets. These observed "predictive tendencies" cannot be solely attributed to image statistics or factors such as radiographic position or patient gender. Instead, they are pathology-specific and linked to higher-level image attributes. The study underscores the importance of diversifying pathology representation to enhance the robustness and generalizability of AI models.

**Strengths:**

- Paper is well written and structured.
- Thorough statistical analysis of the results.
- Dataset-centric approach to investigate robustness and fairness.
- Introduce the concept of “predictive tendency”: models trained on different datasets may demonstrate systematic disparities in predictions, irrespective of their overall performance.

**Weaknesses:**

- The distinction between your study and Gichoya et al. 2022 is not sufficiently elaborated.
Gichoya Judy Wawira, et al. "AI recognition of patient race in medical imaging: a modelling study." The Lancet Digital Health 4.6 (2022): e406-e414.

**Detailed Comments:**

Minor comment:
- I would suggest to present Figure 3 before the results, it would make sense to add a reference to this figure in Section 2.2.
- The sentence seems to be missing a word here “If there was such a correlation, one would expect differences in the intensity histograms between images that have a predictive tendency towards […] and those with predictive tendency towards CXP”.
- The availability of the code would strength the transparency and reproducibility of the results.

**Justification Of The Preliminary Rating:**

This paper addresses an important aspect for deploying robust and fair AI model for chest X-ray classification. They investigate attributes beyond structure and demographics to better understand the models. They introduce the concept of "predictive trends" and emphasize the importance of diversifying pathology representation to enhance the robustness and generalizability of AI models.

**Questions To Address In The Rebuttal:**

1. Highlight differences between your work and Gichoya, Judy Wawira, et al. "AI recognition of patient race in medical imaging: a modelling study." The Lancet Digital Health 4.6 (2022): e406-e414.

**Special Issue:**

Yes

---

> ### Author Response · Authors · 2024-03-17
>
> __Order of Figures__ We thank the reviewer for this practical advice and have moved Figure 3 to the methods section in cross-reference with Section 2.2.
>
> __Incomplete sentence__ We apologize for this oversight and have added the missing word. The complete sentence reads: *“If there were such a correlation, one would expect differences in the intensity histograms between images that have a predictive tendency towards MMC versus those with a predictive tendency towards CXP.”*
>
> __Code availability__ We thank the reviewer for this suggestion. To enhance reproducibility and transparency, we will make our code available on GitHub upon acceptance and added the link to the manuscript: https://github.com/lotterlab/xray_generalization.git
>
> __Differences between our work and Gichoya et al.__ We thank the reviewer for the opportunity to clarify the differences between the work by Gichoya et al. and ours. Gichoya et al. focus on the ability of deep learning (DL) models to predict race from medical images; our work focuses on image-based factors that contribute to performance differences of DL classifiers trained on different datasets. Thus, while both works study AI models in the context of medical image characteristics, they have different goals and focuses. Importantly, we find that our identified predictive tendencies do not correlate with structured attributes such as patient race, age, and sex (Section 3.2 and Appendix D). Beyond these structured attributes, we performed several additional analyses and explainability techniques to understand the image characteristics underlying the predictive tendencies. Among others, we performed an experiment inspired by Gichoya et al. using low and high-pass filters to modify the frequency content of the images. This approach, however, diverges in its application since the AI models are developed for different tasks. We have now clarified these differences in the last paragraph in the Results section.

---

> > ### Comment · Reviewer_ogFB · 2024-03-26
> >
> > Thank you for your response and clarifications, I am satisfied with the replies and have no further comments.

---

### Official Review · Reviewer_ui4s · 2024-02-26

**Confidence:** 3
**Preliminary Rating:** 3
**Final Rating:** 5

**Summary:**

This work adopts a “dataset-centric” perspective to examine performance gaps between DL models trained on different datasets, revealing that these gaps vary across pathologies and are influenced by their representation in the datasets. The key contribution of the work is introducing the concept of “predictive tendency”, which indicates that models trained on different datasets can exhibit systematic differences in predictions regardless of overall performance.

**Strengths:**

1. The key contribution of the work is introducing the concept of “predictive tendency”. There are some performance gaps between different models based on different datasets. The evaluation of this gap may help to maintain the consistency in disease diagnosis.
2. Reveal the gaps between two datasets by statistical distribution.

**Weaknesses:**

1. There is a lack of ablation experiments to the extent to which these influencing factors affect the predictions of the model, such as controlling the effects of age, gender, etc., and observing the prediction results under different disease conditions.
2. Predictive tendency can find the prediction difference between two datasets. However, which factors contribute to this difference are not better quantified. For example, given a CXR image and two prediction results from different models, How much does the age factor contribute to the tendency?

**Detailed Comments:**

1. The image size in the experiment is 224x224, it would be a bit small. The intensity normalization to a range of -1024 to 1024, does this mean the processed image's grayscale ranges from -1024 to 1024?
2. What is the output of the CDM model? Is it predictive tendency or the prediction score of diseases? Please provide more details about CDMs.
3. Provide some specific examples to explain your results may be better.

**Justification Of Final Rating:**

I think this paper gives a unique view of the AI model across different datasets. The predictive tendency is a good concept to evaluate the prediction difference between models trained on different datasets. Based on this idea and the response, I think this paper deserves accepted.

**Justification Of The Preliminary Rating:**

1. The key contribution of the work is introducing the concept of “predictive tendency”. There are some performance gaps between different models based on different datasets. The evaluation of this gap may help to maintain the consistency in disease diagnosis.
2. The introduction to CDM is relatively brief, with too much statistical information making it difficult to grasp the key points of predictive tendency, and a lack of specific examples to support the experimental results.

**Questions To Address In The Rebuttal:**

1. Provide details about CDMs to explain the predictive tendency.
2. Provide some specific examples to explain your results.

---

> ### Author Response · Authors · 2024-03-17
>
> __1. Questions regarding the Comparative Dataset Models (CDM)__ Thank you for the opportunity to clarify aspects of our Comparative Dataset Models (CDMs) and their outputs. The CDMs are trained to predict the predictive tendency, i.e., the relative difference in pathology-specific prediction scores between models trained on different datasets. Thus, the output of the CDM is an estimate of the predictive tendency calculated based on the pathology predictions of the Pathology Prediction Models (PPMs).
> We modified the figure describing the CDMs to more explicitly define their outputs and how they relate to the predictive tendencies. We have also moved this figure to the Methods section to accompany the written description in Section 2.2. We have additionally expanded upon this written description to accompany the mathematical formulation.
>
> __2. Provide examples to explain the results__ Thank you for this suggestion. We have now included example images that received the highest and lowest CDM prediction values for each dataset in Appendix F. The example images align with our other results in that there is large heterogeneity amongst these images. We identified several contexts where particular radiographic views are overrepresented, which are consistent with correlations between radiographic views and the predictive tendencies we describe in Section 3.2.
>
> __3. Factors affecting predictions, such as age and gender__ We appreciate the reviewer's insight on how structured attributes may contribute to our findings. In the original submitted version of the manuscript, we had indeed quantitatively assessed how age, gender, race, and radiographic view may contribute to the predictive tendency. We have now expanded this analysis to explicitly assess if these factors are associated with the CDM predictions. We describe both of these aspects below.
>
> __Predictive tendency__ We quantitatively assessed the association between the predictive tendency and structured attributes using two approaches. First, we assess the statistical significance of the relationships using a Kruskal-Wallis test. Second, we use epsilon squared as a standardized metric to quantify the magnitude of an attribute’s impact on the predictive tendency. These results are described in Section 3.2, Figure 2, and Appendix D (Figures 9, 10, and Table 4). Importantly, we find very low associations between these structured attributes and the predictive tendencies, which we have now expanded upon in Section 3.2.
>
> __Comparative Dataset Models__ Additionally, we have performed a new analysis on the impact of the same set of structured attributes on the CDM predictions (inferred predictive tendencies). These findings largely mirror the effects described in the paper for the predictive tendencies, where these structured attributes can only explain a small fraction of the CDM predictions. Overall, we found that the radiographic view had the strongest associations with the CDM predictions, though the effect sizes (epsilon squared) remained low, with a range between 0.07 and 0.32 across the pathologies. Patient sex had a statistically significant association with the CDM predictions for pneumonia and pleural effusion but with very small effect sizes (epsilon squared of 0.006 and 0.02, respectively). Patient race and age did not have a statistically significant association with the CDM predictions. We have included this new analysis in Appendix G.
>
> While we are not aware of other works studying our defined predictive tendencies, our findings are consistent with prior work studying the effects of structured attributes on pathology prediction performance. Studies by Seyyed-Kalantari et al. [1] and Wu et al. [2] revealed that only a minor portion of chest X-ray classifiers’ performance discrepancies between the CheXpert and MIMIC datasets stem from structured attributes such as age, gender, and race.
>
> __4. Preprocessing of images__ As our goal is to study generalization in the context of standard AI approaches, we adopted best practices for chest X-ray image preprocessing as performed in prior studies. Aligning with methodologies used by works such as Cohen et al. [3] and Pooch et al. [4], we adopt similar image sizes and normalize the grayscale intensity values to a range of -1024 to 1024 to ensure a standardized model input. We have clarified this motivation in Appendix A.
>
> [1] Seyyed-Kalantar et al. (2020). CheXclusion: Fairness gaps in deep chest X-ray classifiers. Biocomputing 2021
> [2] Wu et al. (2021). Explaining medical AI performance disparities across sites with confounder Shapley value analysis. Proceedings of Machine Learning Research.
> [3] Cohen et al. (2020). On the limits of cross-domain generalization in automated X-ray prediction. Proceedings of Machine Learning Research
> [4] Pooch et al. (2020). Can we trust deep learning based diagnosis? The impact of domain shift in chest radiograph classification. International Workshop on Thoracic Image Analysis

---

> > ### Comment · Reviewer_ui4s · 2024-03-26
> >
> > Thank you for your response and clarifications, I am satisfied with the replies and have no further comments.

---

### Official Review · Reviewer_kNys · 2024-03-05

**Confidence:** 5
**Preliminary Rating:** 5
**Recommendation:** Oral
**Final Rating:** 5

**Summary:**

The authors develop a method to analyse classifiers’ discrepancy in term of performances when trained and tested on two different chest X-ray datasets. They introduce the « predictive tendency » metric that compare the ranked outputs of two classifier trained on different data but tested on the same population. This metric allows the author to train a second model that try to predict this metric from an image. They finally analyse the characteristics of images with high and predictable discrepancy.

**Strengths:**

1. Very clear presentation, with insightful and valuable figures
2. The experiments are well thought and extensive: multiple kind of variations are analysed like sex, age and race but also image-based statistics, which is less commonly analysed
3. The reproducibility seems very good
4. The conclusion are very important for the community to assess all kind of bias in deep learning model for medical imaging. While some papers show that sex and race can be linked with model discrepancy [1,2], the authors of the reviewed paper show that it doesn’t explain all the dataset discrepancy and find new way to analyse that.


[1] Laleh Seyyed-Kalantari et al., Underdiagnosis bias of artificial intelligence algorithms applied to chest radiographs in under-served patient populations
[2] Ben Glocker et al., Risk of Bias in Chest Radiography Deep Learning Foundation Models

**Weaknesses:**

No major weakness

The paper might increase a little bit the litterature review. For example this paper [3] do a dataset centric analysis by splitting by sex.

Other paper talked about bias in chest x-ray [2]
[1] Ben Glocker et al., Risk of Bias in Chest Radiography Deep Learning Foundation Models
[3] Agostina J Larrazabal et al, Gender imbalance in medical imaging datasets produces biased classifiers for computer-aided diagnosis

**Detailed Comments:**

na

**Justification Of Final Rating:**

I think that this kind of analyse are very valuable for the community, and this paper does it right. I think that it deserve an Oral presentation given the extensive analysis and the clear presentation

**Justification Of The Preliminary Rating:**

I think that this kind of analyse are very valuable for the community, and this paper does it right.
I think that it deserve an Oral presentation given the extensive analysis and the clear presentation

**Questions To Address In The Rebuttal:**

na

**Special Issue:**

Yes

---

> ### Author Response · Authors · 2024-03-17
>
> We thank the reviewer for their positive review and helpful comments, especially for the observation that the work of Larrazabal et al. represents a data-centric analysis of performance gaps. We have now extended the literature review to include all of the mentioned references as well as additional work in this area, as suggested by the reviewer.

---

> > ### Comment · Reviewer_kNys · 2024-03-27
> >
> > I thanks the authors for their answers. I will take them into account.

---

### Author Response · Authors · 2024-03-17

We sincerely thank all the reviewers for their time and fair and balanced evaluations. We address each reviewer's feedback in a separate comment below the respective review. To ensure clarity and avoid redundancy, we have structured our responses to address comments by subject area and summarize the feedback (bolded) before offering detailed replies. Additionally, citations from the manuscript are highlighted in italics. We look forward to a constructive and engaging discussion during the discussion period.

---

### Comment · Area_Chair_Q4DA · 2024-03-18
**Invitation to reply to authors**

Dear reviewers,

The authors have prepared responses to your comments, which you should now be able to see in OpenReview. We encourage you to reply to their comments, and where necessary, adjust your rating. Please do so before the 27th of March.

---

### Meta-Review · Area_Chair_Q4DA · 2024-04-01

**Recommendation:** Accept (Oral)
**Confidence:** 4

**Metareview:**

The paper studies robustness of models trained on different datasets for the same task and shows systematic differences which cannot be revealed by overall performance, but can be predicted based on pathology and image characteristics. The reviewers agreed on the clarity of the paper and the importance of the investigation. The reviewers had some questions, which appear to have been all addressed in the rebuttal, and the reviewers are in agreement on the quality of the paper. I’m happy to recommend acceptance.

---

### Decision · Program_Chairs · 2024-04-06

Accept (Oral)